# Willingness to pay for antiretroviral therapy, viral load, and premium services; A contingent valuation survey of people living with HIV in southern Nigeria

Olusola Sanwo[1‡]*, Ihoghosa Iyamu[2‡], Augustine Idemudia[3‡], Titilope Badru[1‡], Sylvia Ekponimo[1‡], Dorothy Oqua[4], Olusesan A. Makinde[5], Gambo G. Aliyu[6], Abimbola Kola-Jebutu[3], Jemeh Egwuagu-Pius[3], Chika Obiora-Okafo[3], Moses Bateganya[3], Iorwakwagh Apera[1], Satish Raj Pandey[7], Hadiza Khamofu[1]

1 FHI 360, Guzape, Nigeria, 2 School of Population and Public Health, University of British Columbia, Vancouver, BC, Canada, 3 United States Agency for International Development (USAID), Abuja, Nigeria, 4 Howard University Global Initiative Nigeria, Abuja, Nigeria, 5 Viable Knowledge Masters (VKM), Abuja, Nigeria, 6 National Agency for Control of AIDS (NACA), Abuja, Nigeria, 7 Helen Keller International, Asia Pacific Regional Office, Phnom Penh, Cambodia

‡ OS, II, AI, TB and SE share first authors on this work.
* osanwo@fhi360.org, osanwo@gmail.com

**Data Availability Statement:** All relevant data are within the manuscript and its Supporting Information files.

## Abstract

### Background

With stagnating funding for HIV and AIDS control programs in Nigeria, alternative funding models for antiretroviral therapy (ART) including out of pocket payment are being considered to sustain momentum epidemic control targets. We assessed willingness to pay for ART related services, and factors associated with willingness to pay.

### Methods

Between July and August 2019, we conducted a survey among people living with HIV (PLHIV) on ART in 3 states in southern Nigeria. Randomly sampled respondents on ART for at least 6 months, aged ≥ 18 years, able to communicate in English or pidgin English, and consenting to the survey were enrolled. Respondents were asked if they were willing to pay for clinical consultation, antiretroviral drugs (ARVs), viral load testing services and premium ART services (including fast track services). Respondents indicating willingness to pay for any of these services were asked the maximum amount they were willing to pay using contingent valuation methodology. We assessed the weighted proportions of PLHIV on ART willing to pay for ART and used survey-featured logistic regression measures to assess sociodemographic and ART related factors associated with willingness to pay for ART services.

**Funding:** This study was made possible by the generous support of the American people through the U.S. Agency for International Development (USAID) (AID-620-A-11-00002) and (7200AA19CA00002) awarded to OS, II, AI, TB, SE, DO, SM, JP, SP, and HK.

**Competing interests:** The authors have declared that no competing interests exist.

## Results

Overall, 1,598 PLHIV with a mean age of 39.03 years (standard deviation [SD]: 11.23 years), were included in this analysis. Of these, 65.8% (1,079), 73.9% (1,192), 61.0% (995) and 33.6% (472) were willing to pay for ART consultation, ARVs, viral load testing services and premium ART services respectively. The median maximum amount PLHIV were willing to pay for clinical consultation and for ARVs was NGN1,000 (USD equivalent of $2.78; inter-quartile range [IQR]: 500–2,000) respectively, and NGN2,500 (USD equivalent of $6.94; IQR: NGN1,000–5,000) and NGN2,000 (USD equivalent of $5.56; IQR: NGN1,000–3,000) for viral load testing and premium ART services respectively. Receiving ART in Lagos state, being employed and having a monthly income of NGN100,000 or more was associated with willingness to pay for the various ART services.

## Conclusion

We found generally high-level of willingness to pay for ART consultation, ARVs and viral load testing services but low willingness to pay for premium ART services among PLHIV on ART. The maximum amount PLHIV were willing to pay for various ART services fell short of benchmarks for alternative funding but can potentially supplement ART by funding differenti-ated service delivery models that require nominal amounts to facilitate person-centered dif-ferentiated service delivery models.

## Introduction

Despite significant progress in managing the HIV/AIDS epidemic in Nigeria, the country still has the largest HIV burden in West and Central Africa and the 4[th] largest burden in the world [1, 2]. With over 1.8 million people living with HIV (PLHIV) and about 1.5 million people currently on lifelong antiretroviral therapy (ART), significant healthcare investments are required to foster progress towards the Joint United Nations Programme on HIV/AIDS (UNAIDS) 2025 interim targets for HIV epidemic control [2–4].

Anchored on people-centered, integrated care, the revised UNAIDS targets aim to ensure that 95% of PLHIV know their HIV status, 95% of PLHIV who know their HIV status initiate ART and 95% of PLHIV on ART are virally suppressed, amongst other community and prevention targets [4]. With support from international donors, Nigeria has helped 90% of the estimated people living with HIV to know their HIV status and reduced the prevalence of the disease from 3.2% in 2016 to 1.4% in 2019. The country has facilitated reduction in HIV-related death rates from 2.83% to 2.68% over the past 5 years [5, 6]. However, amidst reduced funding from donors, the country requires the development of plans to maintain momentum towards reaching and maintaining the UNAIDS targets for HIV epidemic control.

Most resources for the HIV/AIDS response have been provided by foreign donors including the US President's Emergency Plan for AIDS Relief (PEPFAR) and the Global Fund to Fight AIDS, Tuberculosis and Malaria (the Global Fund) and these resources have declined over the past 4 years) [5, 6]. The Nigerian government has developed plans to generate local funding to support the response, including the signing of the National Health Insurance Authority Act of 2022 into law. However, concerns remain about its implementation, especially in light of previous failed attempts to secure health for all among some states in the

country [7]. For example, despite previous attempts at securing public funding for universal health coverage, over 90% of health expenditure is funded through out-of-pocket payments (OOP) [8, 9].

The current HIV response provides free ART for Nigerians in designated health facilities, largely through funding from external donors [8, 9], with indirect medical expenses including clinical consultation fees, laboratory services and therapy for opportunistic infections and other comorbidities being covered through out of pocket (OOP) payments or other health insurance schemes [8, 9]. With the dwindling funding, alternative models have been sought including facilitating OOP for aspects of ART [10]. For example, the United States Agency for International Development (USAID) funded the implementation of a private sector initiative called Sustainable Financing Initiative (SFI) under the Strengthening Integrated Delivery of HIV/ AIDS Services (SIDHAS) project implemented by FHI 360. SFI targeted PLHIVs who are willing and able to pay for ART services. While potentially propagating existing health inequities, OOP for ART may be a viable option for supplemental funding of the HIV program [11]. However, this will depend on clients' willingness to pay for HIV services.

Studies in Nigeria and South-East Asia have sought to understand factors associated with willingness to pay for HIV services, aiming to segment the population and offer paid services where appropriate [11–13]. These studies highlighted employment status, income and hospitalization status as important factors associated with willingness to pay for HIV services [11, 12]. However, they inadequately consider contextual factors. For example, they mostly conceptualize ART as a broad group of services and have not deconstructed it into sub-categories including consultation, drugs and ARVs as there is potential to subsidize specific components of ART [11, 13]. Further, majority of these studies have been in single sites which may not be representative of large-scale programs, especially in Nigeria [11].

This study, using data from a representative survey of people living with HIV from a large-scale HIV program in three states in southern Nigeria, sought to assess willingness to pay for ART services, and identify factors associated with willingness to pay using regression methods appropriate for complex survey designs. This analysis can help identify factors to be considered if large-scale HIV programs may leverage out of pocket models to supplement existing HIV care and treatment program funding.

## Methods

### Study design, setting, and population

This was a cross-sectional study of PLHIV, receiving ART in three (3) states in southern Nigeria (Akwa Ibom, Cross River & Lagos). The study used contingent valuation methodology (CVM) which has been commonly used in cost-benefit analyses and to determine what value PLHIVs place on ART and related services [11, 13, 14]. Study sites were those supported by the USAID funded SIDHAS project which was implemented by FHI 360. A cluster sampling design was used to select ninety-eight (98) facilities within the three states providing ART services that met the inclusion criteria using probability proportional to size. The study included public and private health facilities that provide comprehensive HIV/AIDS care and treatment services (ARVs, clinical and laboratory assessments) to PLHIV. These facilities included primary, secondary, and tertiary level facilities located in rural and urban settings. Health facilities included in the study were those with 200 or more PLHIV on ART (for the secondary and tertiary health facilities) and those with 20 or more PLHIV on ART (for primary health care facilities).

We calculated sample size using the Kish-Leslie formula with the following assumptions: 33.3%; proportion of HIV clients who are willing to pay for a monthly supply of ARV supply

[11], a precision of 3.5%, an alpha of 5%. A sample size of 697 was calculated. The sample size was increased by 25%, giving a total of 871 to accommodate for non-response. In addition, a design effect of 2.0 was assumed, giving a minimum sample size of 1,743.

We stratified facilities into public and private facilities and proportionally allocated the sample across these strata. The sampling frame was generated using the ART appointment register and the electronic medical record Lafiya Management Information System (LAMIS). We included clients who were aged 18 years and above, had been on ART for at least 6 months, accessing care (i.e., attending HIV clinic) at the selected facility during the study period, and were able to communicate in either English or Pidgin English. Severely ill patients and PLHIV working as staff or in volunteer roles within the selected facilities were excluded from the study because we expected some social desirability bias from this group. Simple random sampling technique utilizing a random number generator was used to select clients at selected facilities.

## Data collection

Between July and August 2019, interviewers administered the survey questionnaire to consenting eligible clients during their clinic visits using a mobile Open Data Kit (ODK)-based data collection system called SurveyCTO. In addition to assessing respondents' socio-demographic characteristics and duration on ART, the survey elicited respondents' willingness to pay for various HIV and ART related services using a bidding process based on CVM. To assess the amount participants were willing to pay for clinical consultation, ARVs, viral load (VL) tests and premium services which could include fast-track clinic, concierge pharmacy and laboratory services (where laboratory services are delivered to clients within the fast-track clinic), flexible ART scheduling and longer clinic consultation time. The bidding process began with a projected cost for each service and progressed to the maximum amount they were willing to pay (Fig 1). The projected cost was determined based on the contingent valuation methodology for measuring WTP. Among respondents who were unwilling to pay for the projected cost of a particular service, the interviewer elicited the actual amount they were willing to pay.

## Analytical sample and study variables

We included respondents who provided valid responses to survey items related to our main outcomes and the covariates. The main outcome variables were willingness to pay for consultation, ARVs, viral load (VL) assessments and premium ART services including private and fast track services. Outcomes were coded as either yes or no. Additional information on sociodemographic variables including gender (female/male), age (in years), marital status (single, married/cohabiting and widowed/divorced), highest level of education (none, primary, secondary and tertiary), employment status (unemployed, self-employed and employed), monthly household income (less than NGN18,500, NGN18,500–100,000 and greater than NGN100,000), and household role (dependent, sole provider and joint provider) were collected. Health access and ART-related information including having health insurance (yes or no) and duration on ART (12 months or less, 13–24 months and 25 months or more) were also collected.

## Statistical analysis

The data was summarized using simple descriptive statistics including frequencies and proportions of the clients willing to pay for ART services, as well as the median and interquartile ranges (IQR) of the maximum amount participants were willing to pay for the services. The study assessed the US dollar value of the maximum amount PLHIV were willing to pay for ART services by converting at a rate of 1$ to NGN359.98 which was prevalent at the time of

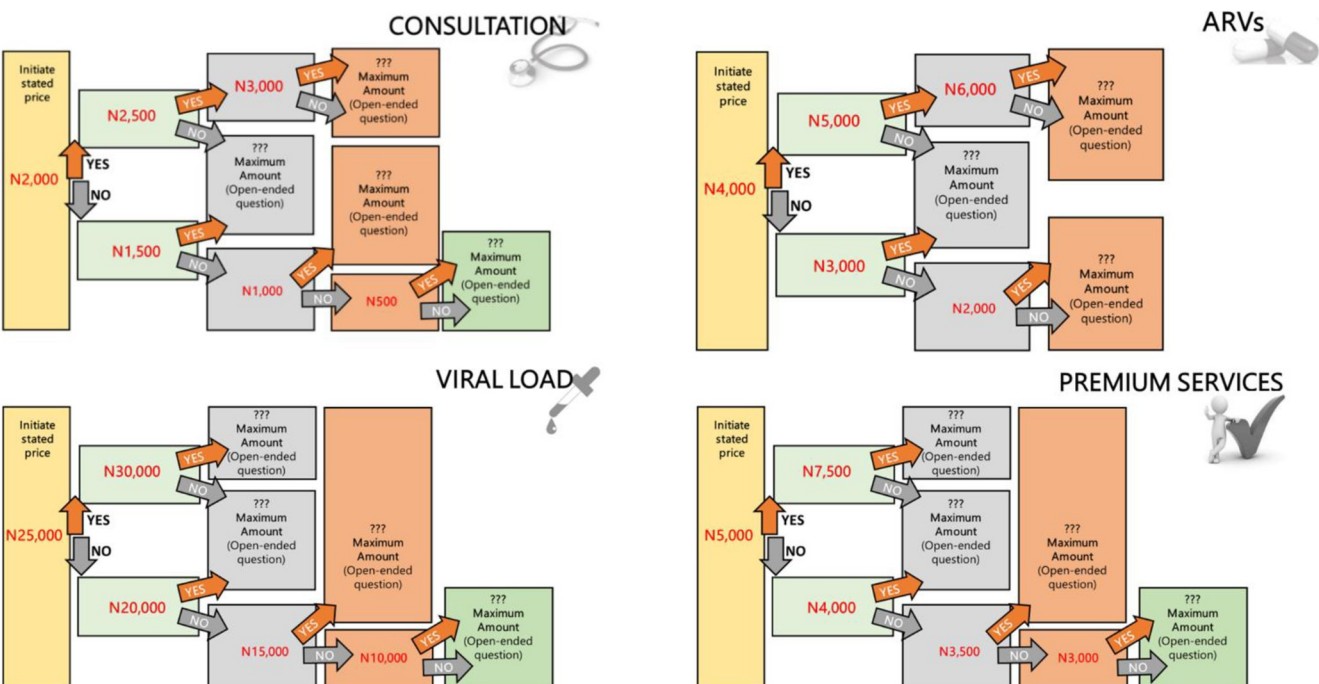

**Fig 1. Illustration of contingent valuation method for eliciting maximum amount willing to pay for consultation, ARVs, viral load and premium services.**

the study. Bivariate analyses were conducted using Thomas-Rao corrections to chi-square test to assess associations between categorical variables and the outcomes of interest [15]. The study used survey-featured multivariable logistic regression models to determine the factors associated with WTP for the various ART-related services and reported the odds ratios (OR) and corresponding 95% confidence intervals (CI). For each outcome, after retaining predictors identified in the literature, we used automated backward elimination method based on Akaike's information criterion (AIC) to select our final models [16]. We assessed collinearity using variance inflation factors (VIF) and set a cut-off at VIF < 10. The models were assessed using Archer-Lemeshow goodness of fit test [17]. Our analyses were conducted using complete cases and were performed using R ver. 4.0.2 [18].

## Ethical considerations

Ethical approval for the study was obtained from the National Health Research Ethics Committee (NHREC), Federal Ministry of Health, Abuja and from FHI 360 Protection of Human Subjects Committee (PHSC) at the FHI 360 head office, Durham NC. Informed consent was taken from all respondents prior to their participation in the study and all study procedures conformed with the principles embodied in the Helsinki Declaration.

## Results

### Sociodemographic and select characteristics of respondents

A total of 1,781 survey responses was collected from SIDHAS clients, representing 80,083 clients in the selected facilities (31,408, 18,551 and 30,124 from Akwa-Ibom, Cross River, and Lagos states respectively). Among these responses, 183 were excluded for missing or invalid responses. Therefore, we included total of 1598 records of respondents whose data were

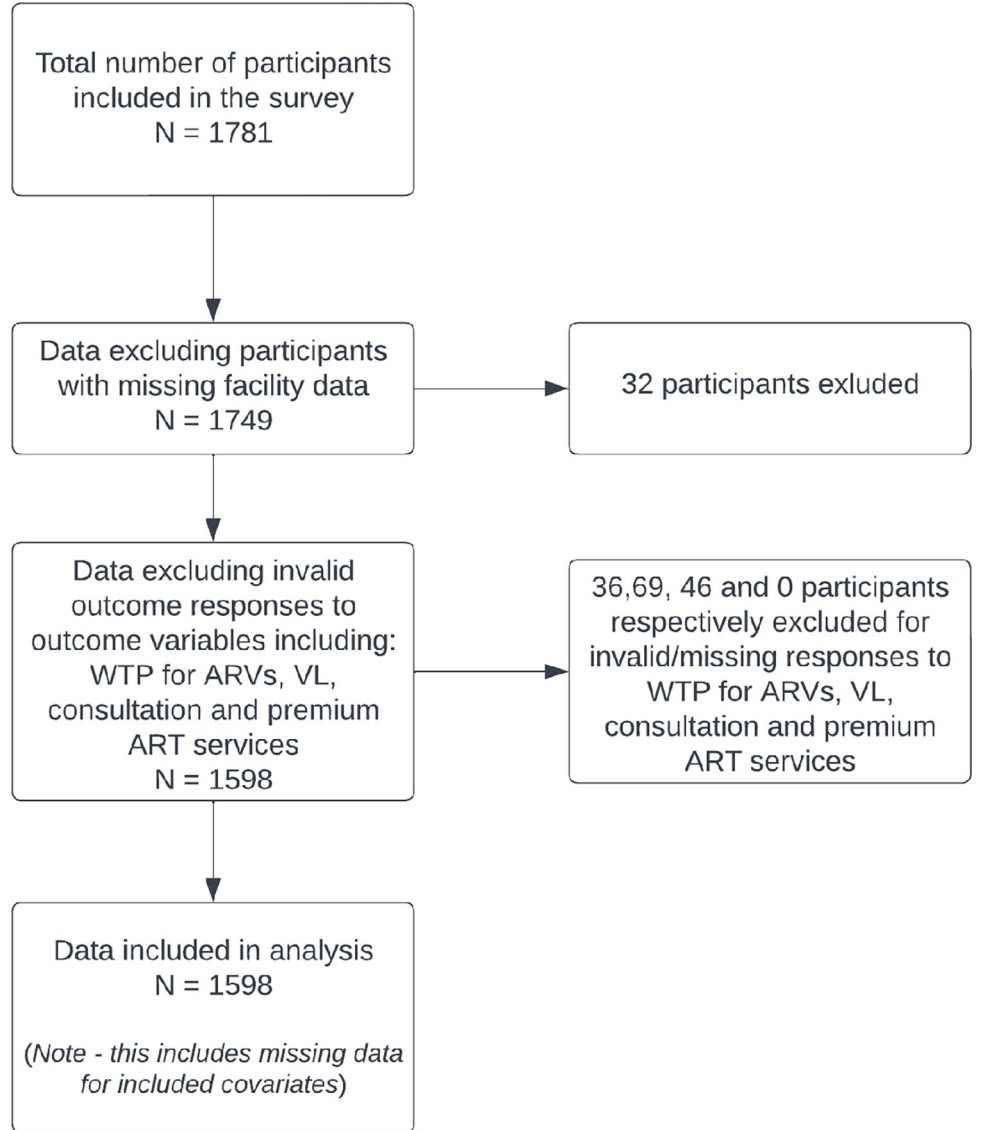

**Fig 2. Flow chart showing inclusion of respondents for the study: Willingness to pay for HIV services survey 2019 —Nigeria.**

analyzed, (Fig 2). Respondents had a mean age of 39.03 years (standard deviation [SD]: 11.23 years) (Table 1). Of these, 1,356 (70.1%) were receiving ART at public health facilities, 727 (48.1%) were on ART in Akwa-Ibom state, 1,117 (68.8%) were female, 836 (52.6%) were married or cohabiting, 831 (51.7%) had secondary level of education, 942 (58.5%) were self-employed, 468 (29.0%) reported no dependents, 801 (49.9%) reported a household income of NGN18,000 or less, 815 (58.7%) had been on ART for at least 25 months and 1534 (96.4%) reported having no form of health insurance.

## Willingness to pay for ART services

Overall, 65.8% (1,079) of PLHIV on ART in the 3 states included in the study were willing to pay for clinical consultation services, 73.9% (1,192) were willing to pay for ARVs, 61.0% (995)

**Table 1. Study sample characteristics and willingness to pay for consultation, ARVs, HIV viral load testing and premium ART services: Willingness to pay for HIV services survey 2019 –Nigeria.**

| Variables | Overall sample | WTP consultation = yes | WTP consultation = no | WTP ARVs = yes | WTP ARVs = no | WTP VL = yes | WTP VL = no | WTP premium = yes | WTP premium = no |
|---|---|---|---|---|---|---|---|---|---|
| | n (%) [a] | n (%) [a] | n (%) [a] | n (%) [a] | n (%) [a] | n (%) [a] | n (%) [a] | n (%) [a] | n (%) [a] |
| Sample size (n) | 1598 (100.0) | 1079 (65.8) | 519 (34.2) | 1192 (73.9) | 406 (26.1) | 995 (61.0) | 603 (39.0) | 472 (33.6) | 1126 (66.4) |
| **State** | | | | | | | | | |
| Akwa-Ibom | 727 (48.1) | 453 (44.3) | 274 (55.5) | 524 (47.2) | 203 (50.7) | 416 (45.5) | 311 (52.3) | 255 (54.5) | 472 (44.8) |
| Cross River | 454 (31.7) | 285 (30.8) | 169 (33.6) | 316 (29.7) | 138 (37.6) | 265 (29.0) | 189 (35.9) | 122 (27.8) | 332 (33.7) |
| Lagos | 417 (20.2) | 341 (25.0) | 76 (10.9) | 352 (23.1) | 65 (11.8) | 314(25.5) | 103 (11.8) | 95 (17.6) | 322 (21.5) |
| **Facility type** | | | | | | | | | |
| Private | 242 (29.9) | 133 (26.6) | 109 (36.3) | 159 (27.8) | 83 (35.8) | 123 (27.2) | 119 (34.1) | 70 (34.0) | 172 (27.8) |
| Public | 1356 (70.1) | 946 (73.4) | 410 (63.7) | 1033 (72.2) | 323 (64.2) | 872 (72.8) | 484 (65.9) | 402 (66.0) | 954 (72.2) |
| **Gender** | | | | | | | | | |
| Female | 1117 (68.8) | 747 (67.6) | 370 (71.2) | 820 (67.4) | 297 (72.7) | 674 (66.9) | 443 (71.9) | 309 (65.1) | 808 (70.7) |
| Male | 481 (31.2) | 332 (32.4) | 149 (28.8) | 372 (32.6) | 109 (27.3) | 321 (33.1) | 160 (28.1) | 163 (34.9) | 318 (29.3) |
| **Age in years (mean (SD))** | 39.03 (11.23) | 38.91 (10.84) | 39.29 (11.94) | 39.01 (10.65) | 39.10 (12.71) | 38.65 (10.92) | 39.66 (11.68) | 39.23 (11.58) | 38.95 (11.03) |
| **Marital status** | | | | | | | | | |
| Single | 467 (28.5) | 287 (24.4) | 180 (36.3) | 330 (27.4) | 137 (31.5) | 281 (27.4) | 186 (30.1) | 126 (26.4) | 341 (29.5) |
| Married/cohabiting | 836 (52.6) | 608 (58.0) | 228 (42.1) | 679 (56.9) | 157 (40.2) | 560 (56.8) | 276 (46.0) | 270 (56.2) | 566 (50.7) |
| Widowed/ Divorced | 295 (19.0) | 184 (17.6) | 111 (21.6) | 183 (15.7) | 112 (28.3) | 154 (15.8) | 141 (23.9) | 76 (17.4) | 219 (19.8) |
| **Duration on antiretroviral therapy** | | | | | | | | | |
| 12 months or less | 362 (18.1) | 253 (18.3) | 109 (17.6) | 265 (17.1) | 97 (20.8) | 232 (17.4) | 130 (19.1) | 112 (19.0) | 250 (17.6) |
| 13–24 months | 421 (23.2) | 291 (24.5) | 130 (20.7) | 318 (24.2) | 103 (20.2) | 274 (24.8) | 147 (20.6) | 129 (26.4) | 292 (21.6) |
| 25 months or more | 815 (58.7) | 535 (57.2) | 280 (61.7) | 609 (58.6) | 206 (59.0) | 489 (57.7) | 326 (60.2) | 231 (54.6) | 584 (60.8) |
| **Highest level of education** | | | | | | | | | |
| None | 83 (5.4) | 48 (4.4) | 35 (7.3) | 53 (4.3) | 30 (8.5) | 41 (3.4) | 42 (8.6) | 11 (3.7) | 72 (6.2) |
| Primary | 374 (23.8) | 250 (23.1) | 124 (25.2) | 270 (23.6) | 104 (24.4) | 222 (22.3) | 152 (26.1) | 104 (22.9) | 270 (24.3) |
| Secondary | 831 (51.7) | 541 (49.9) | 290 (55.2) | 605 (50.5) | 226 (55.3) | 503 (50.8) | 328 (53.2) | 247 (46.9) | 584 (54.2) |
| Tertiary | 310 (19.0) | 240 (22.6) | 70 (12.2) | 264 (21.6) | 46 (11.9) | 229 (23.5) | 81 (12.1) | 110 (26.5) | 200 (15.3) |
| **Employment status** | | | | | | | | | |
| Unemployed | 431 (27.1) | 249 (23.6) | 182 (33.8) | 279 (23.3) | 152 (37.8) | 224 (23.0) | 207 (33.5) | 115 (24.8) | 316 (28.2) |

*(Continued)*

**Table 1.** (Continued)

| Variables | Overall sample | WTP consultation = yes | WTP consultation = no | WTP ARVs = yes | WTP ARVs = no | WTP VL = yes | WTP VL = no | WTP premium = yes | WTP premium = no |
|---|---|---|---|---|---|---|---|---|---|
| | n (%) [a] | n (%) [a] | n (%) [a] | n (%) [a] | n (%) [a] | n (%) [a] | n (%) [a] | n (%) [a] | n (%) [a] |
| Employed | 225 (14.4) | 187 (18.4) | 38 (6.8) | 193 (16.5) | 32 (8.6) | 182 (19.0) | 43 (7.4) | 103 (21.7) | 122 (10.8) |
| Self-employed | 942 (58.5) | 643 (58.0) | 299 (59.4) | 720 (60.2) | 222 (53.6) | 589 (58.0) | 353 (59.1) | 254 (53.5) | 688 (61.0) |
| **Health insurance** | | | | | | | | | |
| No | 1534 (96.4) | 1031 (95.8) | 503 (97.6) | 1141 (96.2) | 393 (97.1) | 950 (96.0) | 584 (97.1) | 442 (93.1) | 1092 (98.1) |
| Yes | 64 (3.6) | 48 (4.2) | 16 (2.4) | 51 (3.8) | 13 (2.9) | 45 (4.0) | 19 (2.9) | 30 (6.9) | 34 (1.9) |
| **No of dependents** | | | | | | | | | |
| None | 468 (29.0) | 271 (24.4) | 197 (37.8) | 311 (24.8) | 157 (40.7) | 277 (26.2) | 191 (33.3) | 106 (22.0) | 362 (32.5) |
| 1–2 | 587 (34.3) | 412 (36.4) | 175 (30.4) | 436 (36.6) | 151 (28.0) | 355 (35.9) | 232 (31.9) | 178 (39.5) | 409 (31.7) |
| 3 and above | 543 (36.7) | 396 (39.2) | 147 (31.8) | 445 (38.6) | 98 (31.3) | 363 (37.9) | 180 (34.8) | 188 (38.5) | 355 (35.8) |
| **Household monthly income (NGN)** | | | | | | | | | |
| < 18,500 | 801 (49.9) | 460 (43.1) | 341 (63.2) | 530 (45.2) | 271 (63.5) | 428 (43.3) | 373 (60.4) | 229 (42.9) | 572 (53.5) |
| 18,500–100,000 | 735 (44.7) | 566 (49.9) | 169 (34.8) | 605 (48.5) | 130 (33.9) | 516 (49.2) | 219 (37.6) | 214 (48.0) | 521 (43.0) |
| >100,000 | 62 (5.3) | 53 (7.1) | 9 (2.0) | 57 (6.3) | 5 (2.6) | 51 (7.5) | 11 (1.9) | 29 (9.1) | 33 (3.5) |
| **Role in household** | | | | | | | | | |
| Dependent | 353 (20.2) | 198 (17.5) | 155 (25.5) | 216 (16.7) | 137 (30.3) | 174 (16.0) | 179 (26.8) | 66 (14.3) | 287 (23.3) |
| Sole provider | 652 (42.5) | 437 (40.8) | 215 (45.8) | 482 (42.6) | 170 (42.4) | 413 (42.2) | 239 (43.1) | 204 (42.1) | 448 (42.8) |
| Joint provider | 593 (37.2) | 444 (41.6) | 149 (37.2) | 494 (40.7) | 99 (27.3) | 408 (41.8) | 185 (30.0) | 202 (43.6) | 391 (34.0) |

WTP: willingness to pay; ARVs: antiretroviral drugs; VL: viral load testing; ART: antiretroviral therapy; Decimal points may not add up to 100% due to rounding error. [c] All proportions are weighted to the population of people living with HIV in care, using sampling weights, sampling units and strata available in the survey data.

were willing to pay for viral load testing and 33.6% (472) were willing to pay for premium ART services. The median maximum amount PLHIV were willing to pay for ART consultation and for ARVs was NGN1,000 (USD equivalent—$2.78, interquartile range [IQR]: 500–2,000) respectively (Fig 3). Further, the median maximum amount PLHIV were willing to pay per viral load test and premium ART services was NGN2,500 (USD equivalent—$6.94, IQR: NGN1,000–5,000) and NGN2,000 (USD equivalent—$5.56, IQR: NGN1,000–3,000) respectively.

## Factors associated with willingness to pay for ART services

In bivariable analyses, receiving ART in Lagos state (vs. Akwa-Ibom state, odds ratio [OR]: 2.86, 95% confidence interval [CI]: 1.39–5.88), tertiary education (vs. no education, OR: 3.03, 95% CI: 1.10–8.30), being employed (vs. unemployed, OR:3.87, 95%CI: 2.12–7.07), being self-employed (vs. unemployed, OR: 1.40, 95%CI: 1.02–1.92), having monthly household income of NGN100,000 and above (vs. NGN18,000 or less, OR: 5.16, 95%CI: 1.83–14.56) were

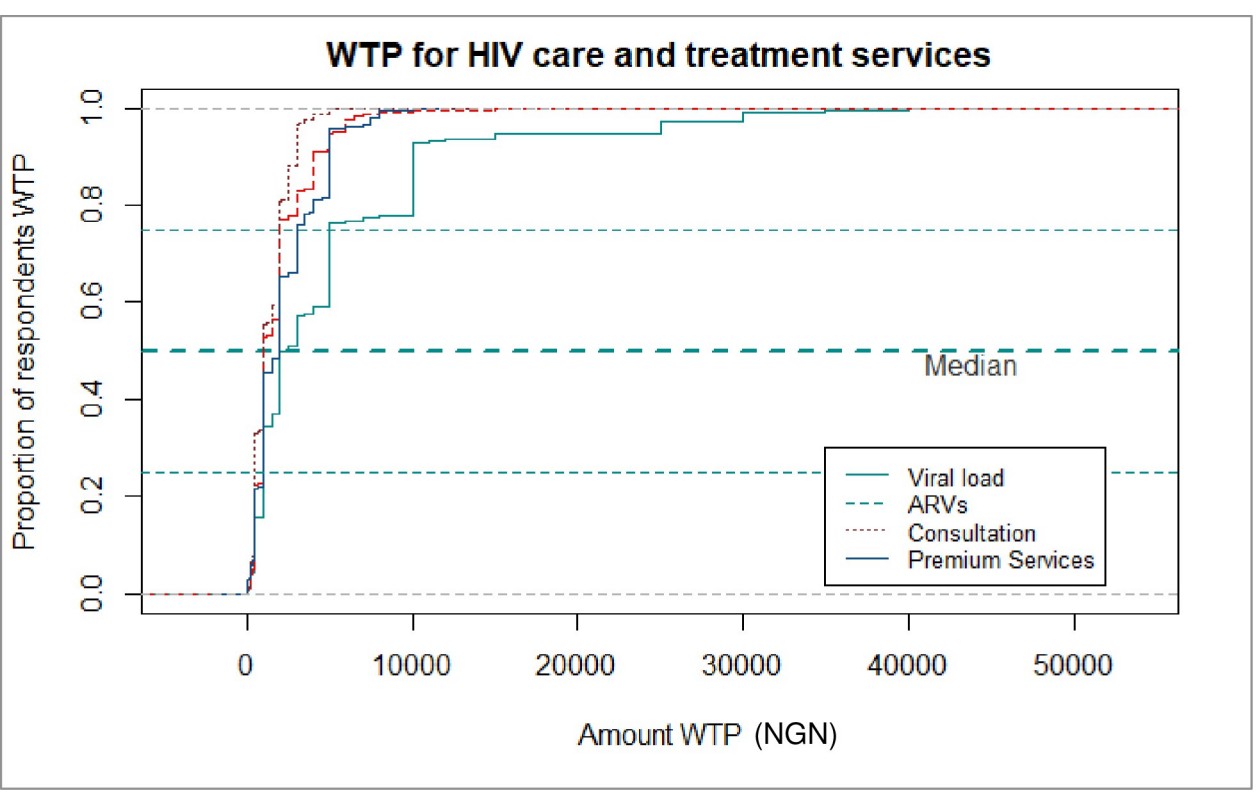

**Fig 3. Cumulative plots of amounts PLHIV are willing to pay for various HIV treatment and care services: Willingness to pay for HIV services survey 2019 –Nigeria.** WTP: Willing to pay; NGN: Nigerian Naira; ARVs: Antiretroviral drugs.

associated with willingness to pay for clinical consulting services (Table 2). Further, receiving ART in Lagos state (vs. Akwa-Ibom state, OR: 2.11, 95%CI: 1.19–3.73), receiving ART in a public health facility (vs. private, OR: 1.44, 95%CI: 1.05–1.99), being married or cohabiting (vs. single, OR: 1.63, 95%CI: 1.23–2.17), tertiary education (vs. no education, OR: 3.57, 95% CI: 1.10–11.51), being employed (vs. unemployed, OR:3.11, 95%CI:1.73–5.60), being self-employed (vs. unemployed, OR: 1.82, 95%CI: 1.26–2.62), having monthly household income of NGN100,000 and above (vs. NGN18,000 or less, OR: 3.43, 95%CI: 1.10–10.68) were associated with willingness to purchase ARVs. Receiving ART in Lagos state (vs. Akwa-Ibom state, OR: 2.48, 95%CI: 1.29–4.77), tertiary education (vs. no education, OR: 4.99, 95% CI: 2.08–11.96), being employed (vs. unemployed, OR:3.77, 95%CI:1.96–7.25), and having monthly household income of NGN100,000 and above (vs. NGN18,000 or less, OR: 5.40, 95%CI: 1.79–16.25) were associated with willingness to pay for viral load testing. Finally, being employed (vs. unemployed, OR:2.29, 95%CI:1.48–3.55), having health insurance (vs. no insurance, OR:3.82, 95%CI:1.90–7.67), and having monthly household income of NGN100,000 and above (vs. NGN18,000 or less, OR: 3.27, 95%CI: 1.78–6.01) were associated with willingness to pay for premium ART services.

In multivariable analyses, receiving ART in Lagos state (vs. Akwa-Ibom state, adjusted odds ratio [aOR]: 2.28, 95%CI: 1.10–4.73), being married or cohabiting (vs. single, aOR: 1.88, 95% CI: 1.14–3.11), being employed (vs. unemployed, aOR: 2.35, 95%CI: 1.15–4.83) and household income of NGN100,000 and above (vs. NGN18,000 or less, aOR: 3.10, 95%CI: 1.04–9.22), were associated with willingness to pay for clinical consultation (Table 3). Being divorced or widowed (vs. single, aOR: 0.52, 95%CI: 0.35–0.79) and being self-employed (vs. unemployed,

**Table 2. Estimates from bivariable logistic regression assessing factors associated with willingness to pay for HIV consultation, ARVs, HIV viral load testing and premium ART services: Willingness to pay for HIV services survey 2019 –Nigeria.**

| Variables | WTP consultation OR (95% CI) [c] | WTP ARVs OR (95% CI) [c] | WTP VL OR (95% CI) [c] | WTP premium OR (95% CI) [c] |
|---|---|---|---|---|
| **State** | | | | |
| Akwa-Ibom | Ref | Ref | Ref | Ref |
| Cross River | 1.15 (0.85, 1.55) | 0.85 (0.60, 1.19) | 0.93 (0.64, 1.35) | 0.68 (0.38, 1.21) |
| Lagos | 2.86 (1.39, 5.88) [a] | 2.11 (1.19, 3.73) [b] | 2.48 (1.29, 4.77) [a] | 0.68 (0.31, 1.45) |
| **Facility type** | | | | |
| Private | Ref | Ref | Ref | Ref |
| Public | 1.58 (1.09, 2.28) [b] | 1.44 (1.05, 1.99) [b] | 1.38 (0.86, 2.22) | 0.75 (0.41, 1.35) |
| **Gender** | | | | |
| Female | Ref | Ref | Ref | Ref |
| Male | 1.19 (0.89, 1.59) | 1.29 (0.89, 1.87) | 1.27 (0.92, 1.73) | 1.29 (0.98, 1.69) |
| **Age in years** | 1.00 (0.98, 1.02) | 1.00 (0.97, 1.02) | 0.99 (0.97, 1.01) | 1.01 (0.99, 0.35) |
| **Marital status** | | | | |
| Single | Ref | Ref | Ref | Ref |
| Married/cohabiting | 2.05 (1.46, 2.88) | 1.63 (1.23, 2.17) [a] | 1.36 (0.92, 2.00) | 1.24 (0.93, 1.66) |
| Widowed/Divorced | 1.21 (0.71, 2.08) | 0.64 (0.40, 1.01) | 0.73 (0.45, 1.17) | 0.98 (0.67, 1.44) |
| **Duration on antiretroviral therapy** | | | | |
| 12 months or less | Ref | Ref | Ref | Ref |
| 13–24 months | 1.14 (0.66, 1.96) | 1.45 (0.78, 2.70) | 1.32 (0.75, 2.31) | 1.14 (0.66, 1.95) |
| 25 months or more | 0.89 (0.61, 1.30) | 1.21 (0.79, 1.85) | 1.05 (0.65, 1.70) | 0.83 (0.54, 1.28) |
| **Highest level of education** | | | | |
| None | Ref | Ref | Ref | Ref |
| Primary | 1.50 (0.83, 2.71) | 1.90 (0.95, 3.78) | 2.19 (1.18, 4.06) [b] | 1.57 (0.51, 4.86) |
| Secondary | 1.48 (0.71, 3.08) | 1.79 (0.78, 4.12) | 2.45 (1.29, 4.65) [a] | 1.44 (0.50, 4.12) |
| Tertiary | 3.03 (1.10, 8.30) [b] | 3.57 (1.10, 11.51) [b] | 4.99 (2.08, 11.96) [a] | 2.89 (0.77, 10.78) |
| **Employment status** | | | | |
| Unemployed | Ref | Ref | Ref | Ref |
| Employed | 3.87 (2.12, 7.07) [a] | 3.11 (1.73, 5.60) [a] | 3.77 (1.96, 7.25) [a] | 2.29 (1.48, 3.55) [a] |
| Self-employed | 1.40 (1.02, 1.92) [a] | 1.82 (1.26, 2.62) [a] | 1.43 (1.05, 1.96) [b] | 1.00 (0.61, 1.64) |
| **Health insurance** | | | | |
| No | Ref | Ref | Ref | Ref |
| Yes | 1.82 (0.65, 5.06) | 1.35 (0.48, 3.81) | 1.43 (0.65, 3.11) | 3.82 (1.90, 7.67) [a] |
| **No of dependents** | | | | |
| None | Ref | Ref | Ref | Ref |
| 1–2 | 1.86 (1.15, 2.99) [b] | 2.14 (1.31, 3.48) [a] | 1.43 (0.90, 2.26) | 1.84 (1.09, 3.12) [b] |
| 3 and above | 1.92 (1.23, 2.98) [a] | 2.01 (1.31, 3.10) [a] | 1.38 (0.83, 2.28) | 1.59 (1.10, 2.30) [b] |
| **Household monthly income (NGN)** | | | | |
| < 18,500 | Ref | Ref | Ref | Ref |
| 18,500–100,000 | 2.10 (1.36, 3.25) [a] | 2.01 (1.26, 3.20) [a] | 1.83 (1.15, 2.89) [b] | 1.39 (0.80, 2.40) |
| >100,000 | 5.16 (1.83, 14.56) [a] | 3.43 (1.10, 10.68) [b] | 5.40 (1.79, 16.25) [a] | 3.27 (1.78, 6.01) [a] |
| **Role in household** | | | | |
| Dependent | Ref | Ref | Ref | Ref |
| Sole provider | 1.29 (1.00, 1.68) | 1.83 (1.31, 2.55) [a] | 1.64 (1.23, 2.19) [a] | 1.61 (1.00, 2.58) |
| Joint provider | 2.11 (1.38, 3.2) [a] | 2.72 (1.85, 4.00) [a] | 2.33 (1.66, 3.27) [a] | 2.09 (1.34, 3.27) [a] |

[a] Significant at $p<0.01$

[b] Significant at $p<0.05$. OR: odds ratio, CI: confidence interval. WTP: willingness to pay; ARVs: antiretroviral drugs; VL: viral load testing.

[c] All estimates (OR, CI) are weighted using sampling weights, sampling units and strata available in the data.

**Table 3. Estimates from multivariable logistic regression assessing factors associated with willingness to pay for HIV consultation, ARVs, HIV viral load testing and premium ART services: Willingness to pay for HIV services survey 2019 –Nigeria.**

| Variables | WTP consultation OR (95% CI) [c, d] | WTP ARVs OR (95% CI) [c, e] | WTP VL OR (95% CI) [c, f] | WTP premium OR (95% CI) [c, g, b] |
|---|---|---|---|---|
| **State** | | | | |
| Akwa-Ibom | Ref | Ref | Ref | Ref |
| Cross River | 1.05 (0.76, 2.98) | 0.80 (0.51, 1.26) | 0.84 (0.57, 1.23) | 0.55 (0.34, 0.90) [b] |
| Lagos | 2.28 (1.10, 4.73) [b] | 1.52 (0.81, 2.86) | 2.06 (1.04, 4.06) [b] | 0.58 (0.26, 1.29) |
| **Facility type** | | | | |
| Private | | Ref | | Ref |
| Public | | 1.22 (0.87, 1.71) | | 0.69 (0.40, 1.22) |
| **Gender** | | | | |
| Female | Ref | Ref | Ref | Ref |
| Male | 1.00 (0.73, 1.37) | 0.94 (0.59, 1.48) | 1.03 (0.70, 1.53) | 1.09 (0.79, 1.51) |
| **Age in years** | 0.98 (0.96, 1.00) [b] | 0.99 (0.97, 1.01) | 0.99 (0.97, 1.00) | 1.00 (0.99, 1.02) |
| **Marital status** | | | | |
| Single | Ref | Ref | | |
| Married/cohabiting | 1.88 (1.14, 3.11) [b] | 1.05 (0.62, 1.77) | | |
| Widowed/Divorced | 1.62 (0.99, 2.66) | 0.52 (0.35, 0.79) [a] | | |
| **Duration on antiretroviral therapy** | | | | |
| 12 months or less | Ref | Ref | Ref | Ref |
| 13–24 months | 1.14 (0.66, 1.96) | 1.59 (0.81, 3.14) | 1.36 (0.73, 2.53) | 1.02 (0.61, 1.70) |
| 25 months or more | 0.89 (0.61, 1.30) | 1.21 (0.83, 1.77) | 1.01 (0.63, 1.61) | 0.68 (0.45, 1.05) |
| **Highest level of education** | | | | |
| None | Ref | Ref | Ref | |
| Primary | 1.20 (0.63, 2.30) | 1.42 (0.69, 2.93) | 1.94 (1.06, 3.54) [b] | |
| Secondary | 1.09 (0.52, 2.27) | 1.18 (0.49, 2.86) | 1.87 (0.93, 3.75) | |
| Tertiary | 1.58 (0.55, 4.53) | 1.82 (0.54, 6.12) | 2.65 (1.01, 6.93) | |
| **Employment status** | | | | |
| Unemployed | Ref | Ref | Ref | Ref |
| Employed | 2.35 (1.15, 4.83) [b] | 1.91 (0.88, 4.13) | 2.56 (1.22, 5.38) [b] | 1.39 (0.81, 2.37) |
| Self-employed | 1.10 (0.84, 1.43) | 1.57 (1.08, 2.28) [b] | 1.31 (1.00, 1.70) | 0.93 (0.60, 1.45) |
| **Health insurance** | | | | |
| No | | | | Ref |
| Yes | | | | 2.17 (0.97, 4.85) |
| **No of dependents** | | | | |
| None | Ref | Ref | Ref | Ref |
| 1–2 | 1.70 (0.97, 2.98) | 2.11 (1.22, 3.67) [a] | 1.43 (0.89, 2.32) | 2.05 (1.10, 3.83) [b] |
| 3 and above | 1.53 (0.89, 2.63) | 1.94 (1.13, 3.32) [b] | 1.32 (0.79, 2.20) | 1.63 (1.03, 2.58) [b] |
| **Household monthly income (NGN)** | | | | |
| < 18,500 | Ref | Ref | Ref | |
| 18,500–100,000 | 1.52 (0.93, 2.47) | 1.58 (0.87, 2.87) | 1.41 (0.78, 2.54) | |
| >100,000 | 3.10 (1.04, 9.22) [b] | 2.22 (0.65, 7.55) | 3.47 (1.03, 11.69) [b] | |
| **Covariate interactions** | | | | |
| Highest level of education by Household income | | | | 1.23 (0.32, 4.76) |
| No education (>NGN100,000 vs <NGN18,500) | | | | |
| No education (NGN18,500–100,000 vs <NGN18,500) | | | | 1.55 (0.38, 6.39) |
| Primary education (>NGN100,000 vs <NGN18,500) | | | | 7.75 (1.12, 53.62) [b] |

*(Continued)*

**Table 3.** (Continued)

| Variables | WTP consultation OR (95% CI) [c, d] | WTP ARVs OR (95% CI) [c, e] | WTP VL OR (95% CI) [c, f] | WTP premium OR (95% CI) [c, g, h] |
|---|---|---|---|---|
| Primary education (NGN18,500–100,000 vs <NGN18,500) | | | | 0.71 (0.36, 1.38) |
| Secondary education (>NGN100,000 vs <NGN18,500) | | | | 12.33 (2.27, 66.99) [a] |
| Secondary education (NGN18,500–100,000 vs <NGN18,500) | | | | 1.67 (0.85, 3.26) |
| Tertiary education (NGN18,500–100,000 vs <NGN18,500) | | | | 1.88 (0.77, 4.61) |

[a] Significant at $p<0.01$

[b] Significant at $p<0.05$. OR: odds ratio, CI: confidence interval. WTP: willingness to pay; ARVs: antiretroviral drugs; VL: viral load testing.

[c] All estimates (OR, CI) are weighted using sampling weights, sampling units and strata available in the data.

[d] Archer-Lemeshow Goodness-of-fit, $p = 0.245$.

[e] Archer-Lemeshow Goodness-of-fit, $p = 0.388$.

[f] Archer-Lemeshow Goodness-of-fit, $p = 0.256$.

[g] Model includes interaction term between education and monthly household income

[h] Archer-Lemeshow Goodness-of-fit, $p = 0.897$. All variance inflating factors (VIF) for all models < 4.

aOR: 1.57, 95%CI: 1.08–2.28) were associated with willingness to pay for ARVs, while primary education (vs. no education, aOR: 1.94, 95%CI: 1.06–3.54), being employed (vs. unemployed, aOR: 1.57, 95%CI: 1.08–2.28), and household income of NGN100,000 and above (vs. NGN18,000 or less, OR: 3.47, 95%CI: 1.03–11.69) were associated with willingness to pay for viral load testing. Contrastingly, we found varied associations between education and willingness to pay for premium ART services based on monthly household income. Among PLHIV with primary education, household income of NGN100,000 and above (vs. NGN18,000 or less, aOR: 7.75, 95%CI: 1.12–53.62) and among PLHIVs with secondary education, household income of NGN100,000 and above (vs. NGN18,000 or less, aOR: 12.33, 95%CI: 2.27–66.99) were associated with willingness to pay for premium ART services.

## Discussion

In this study, majority of the clients in the three states were willing to pay for clinical consultation, to purchase ARVs and pay for viral load services, but only 33.6% were willing to pay for premium services. These results are important as Nigeria explores alternative funding source for the HIV response. We also found that clients who were employed, with monthly household income of NGN100,000 ($277.79) and above, cohabiting or married and on ART in Lagos had higher odds of being willing to pay for ART consultation and viral load testing. Clients who were self-employed had higher odds of being willing to pay for ARVs. Clients who had 1 or more dependents had higher odds of being willing to pay for ART, while those on ART in Cross River state had lower odds of being willing to pay for ART compared with those in Akwa-Ibom state. Further, we identified covariate interaction effects with clients who had primary level of education and monthly household income of NGN100,000 and above having 7 times the odds of being willing to pay for premium ART services while those with secondary level of education and monthly household income of NGN100,000 and above had 12 times the odds of being willing to pay for premium ART services.

Findings of our study concur with other studies that describe general willingness to pay for ART services [11–13]. However, in our study, the proportion of clients willing to pay for ART

was less than suggested in other similar studies in Vietnam where the proportion of clients willing to pay for ART was as high as 90% [12, 13]. While the high level of willingness to pay for ART may be in keeping with the cultural acceptance of out of pocket payment models in the country [8, 19], the lower levels of willingness to pay compared to other jurisdictions may reflect the economic situation of clients in Nigeria, which may not be sufficient to sustain these out of pocket payments [5]. With respect to premium ART services, only 33.6% of clients were willing to pay. The median amount clients were willing to pay for the cost of ARV services was less than documented in other studies in Nigeria and in Vietnam [11–13]. One possibility for the lower amounts clients were willing to pay could be the significantly higher NGN to USD rates at the time of our study compared with previous similar studies. This exchange rate has further increased given the economic challenges that Nigeria has experienced over the past 5 years [5]. This study also corroborates other studies that suggest that individuals that are employed, with higher monthly household income and levels of education and with 1–2 dependents (compared to none), have higher odds of being willing to pay for ART services. However, our study extends the literature by identifying different effects of education on willingness to pay for premium ART services based on income levels. Higher odds of being willing to pay were seen among educated people with higher income compared to similarly educated individuals with lower monthly household income.

These findings have implications for the HIV program in Nigeria. To provide local context, as at 2021, the average cost of a specialist consultation across SFI supported facilities was NGN2,000 (the cost of a month's supply of TLD was $4.85 while the cost of providing viral load services at an SFI supported PCR laboratory as at 2019 was NGN25,000 ($69.45). First, we acknowledge that out of pocket models are not sustainable as an overarching strategy and our findings further demonstrate the non-viability of this strategy given the maximum amount clients were willing to pay for ART services [9, 20]. However, the nominal amounts clients are willing to pay for ART may be worth exploring in funding other service delivery models in line with person-centered differentiated service delivery models. For example, other studies have shown that clients using paid ART services involving community pharmacists have similar retention rates and higher viral suppression rates than those using standards of care [21]. Paid services may be offered among various options for people at various socio-economic strata with the aim of freeing up standards of care and improving clinical services wherever clients receive care [11, 21]. However, further research will be required to understand the long-term willingness to pay for ART services after clients have engaged in this process, perhaps as pilot studies. This is important because hypothetical situations presented in a contingent valuation method may not resonate with respondents' perception of their realities [11]. A health equity lens is also important in considering how offering such a variety of paid and free services may have unintended consequences for all clients in care [22].

This is a representative study of clients accessing ART in a large HIV prevention, care, and treatment program within the 3 selected states of Nigeria. Using probability sampling methods allows us draw conclusions about clients' willingness to pay for ART services within the states and can inform program managers about the feasibility of implementing alternative funding models within the states. Further, the study provides further insights about clients' willingness to pay for various aspects of ART. This approach allows us to make recommendations about specific services that might be considered in alternative funding models that may require out of pocket payments. However, we are cautious about the conclusions made from our study given inherent limitations with our methods. First, there was risk of social desirability bias among respondents who may have been more likely to be willing to pay when asked by people perceived to be part of the funding organization. We attempted to limit this by engaging independent data collectors. Further, our studies may have been exposed to unmeasured

confounding given that we did not explicitly explore characteristics of respondents' ART including number and recency of hospitalization episodes, quality of life factors and past experience of catastrophic health spending which may have influenced willingness to pay for ART services [9, 13]. However, we accounted for respondents' economic situation using multiple factors including monthly household income, employment status and number of dependents. As this study was expected to identify possible areas whose costs may be transferred from the donor to the clients, questions were only asked about willingness to pay for the different Anti-retroviral therapy (ART) service delivery areas. Additional studies may go further to explore the willingness to pay for the comprehensive ART services.

## Conclusion

In conclusion, the study found a high-level of willingness to pay for ART consultation, ARVs and viral load testing services but low willingness to pay for premium ART services among clients on ART in Akwa-Ibom, Cross Rivers, and Lagos States in southern Nigeria. The maximum amount clients were willing to pay for various components of ART services fell short of benchmarks for alternative funding for HIV/AIDS treatment programs but can potentially supplement ART by funding differentiated service delivery models that require nominal amounts to facilitate person-centered differentiated service delivery models, while freeing up standards of care to focus limited resources on clients. Further research is however required to understand the long-term willingness to pay for ART services, and associated outcomes including retention in treatment programs after clients have engaged in paid services over longer time periods.

## Supporting information

**S1 Data.**
(XLSX)

## Acknowledgments

The contents are the responsibility of the authors and do not necessarily reflect the views of Family Health International (FHI 360) and USAID.

The authors express special gratitude to Ministries of Health in the three states where the study was implemented—Akwa Ibom, Cross River and Lagos, and the health workers in the facilities who facilitated the entry of research assistants. We are also grateful to Network of People Living with HIV/AIDS in Nigeria (NEPWHAN) and Association of Women Living with HIV/AIDS in Nigeria (ASHWAN) for their supervisory roles during data collection. Finally, we are most grateful to all the clients who participated in the study.

## Author Contributions

**Conceptualization:** Olusola Sanwo, Augustine Idemudia, Sylvia Ekponimo, Jemeh Egwuagu-Pius.

**Data curation:** Olusesan A. Makinde.

**Formal analysis:** Olusola Sanwo, Ihoghosa Iyamu, Augustine Idemudia, Titilope Badru, Olusesan A. Makinde.

**Investigation:** Olusola Sanwo, Augustine Idemudia.

Willingness to pay for antiretroviral therapy: A survey of people living with HIV in southern Nigeria

**Methodology:** Olusola Sanwo, Ihoghosa Iyamu, Augustine Idemudia, Titilope Badru, Sylvia Ekponimo.

**Project administration:** Olusola Sanwo, Satish Raj Pandey, Hadiza Khamofu.

**Resources:** Olusola Sanwo, Satish Raj Pandey, Hadiza Khamofu.

**Software:** Olusola Sanwo.

**Supervision:** Olusola Sanwo, Gambo G. Aliyu, Iorwakwagh Apera, Satish Raj Pandey, Hadiza Khamofu.

**Validation:** Olusola Sanwo, Augustine Idemudia, Hadiza Khamofu.

**Visualization:** Olusola Sanwo, Ihoghosa Iyamu, Augustine Idemudia.

**Writing – original draft:** Olusola Sanwo, Ihoghosa Iyamu, Augustine Idemudia.

**Writing – review & editing:** Olusola Sanwo, Ihoghosa Iyamu, Augustine Idemudia, Titilope Badru, Dorothy Oqua, Olusesan A. Makinde, Abimbola Kola-Jebutu, Jemeh Egwuagu-Pius, Chika Obiora-Okafo, Moses Bateganya, Satish Raj Pandey, Hadiza Khamofu.

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
