## [Decision Letter · Decision Letter 0]

6 Jun 2023

PONE-D-23-14068Willingness to pay for antiretroviral therapy, viral load and premium services; A contingent valuation survey of people living with HIV in southern NigeriaPLOS ONE

Dear Dr. Sanwo,

Thank you for submitting your manuscript to PLOS ONE. After careful consideration, we feel that it has merit but does not fully meet PLOS ONE’s publication criteria as it currently stands. Therefore, we invite you to submit a revised version of the manuscript that addresses the points raised during the review process.

ACADEMIC EDITOR:Thank you for the manuscript. Please address the following and resubmit revised version for review: 1. Ensure to align placement of your tables and figures with plosone guidelines for figures and tables. Submission Guidelines | PLOS ONE2. provide response to individual comments submitted by the 2 reviewers3. Review references and ensure all references are listed. Unlisted references should either be deleted or added.

We look forward to receiving your revised manuscript.

Kind regards,

Ibrahim Jahun, MD, MSC, PhD

Academic Editor

PLOS ONE

Journal Requirements:

2. Please clarify the number of Figures uploaded in your manuscript and PDF file. 

Reviewers' comments:

Reviewer's Responses to Questions

**Comments to the Author**

1. Is the manuscript technically sound, and do the data support the conclusions?

Reviewer #1: Yes

Reviewer #2: Yes

2. Has the statistical analysis been performed appropriately and rigorously? 

Reviewer #1: Yes

Reviewer #2: Yes

3. Have the authors made all data underlying the findings in their manuscript fully available?

Reviewer #1: Yes

Reviewer #2: Yes

4. Is the manuscript presented in an intelligible fashion and written in standard English?

Reviewer #1: No

Reviewer #2: Yes

5. Review Comments to the Author

Reviewer #1: Thank you for submitting this article for publication. The findings add to the peer-review literature and are timely.

Please have a copy editor review the paper. There are incomplete sentences and misspelled words throughout the paper and figures. I suggest adding place and date to figure and table titles.

Readers not familiar with Naira could benefit from providing the exchange rate relative to more commonly traded currencies, i.e., dollar, euro, or pound. Authors should be consistent with currency used. In one section both Naira and dollars were used.

Reviewer #2: The manuscript is technically sound requiring minor edits as outlined below. Additionally, the authors should adhere to PLOS ONE guidelines on tables and figures.

Lines 80 – 2: The goal is not to only attain UNAIDS targets but to attain HIV epidemic control. Consider adding this caveat.

Lines 146 – 7: pharmacy, lab services and longer consultation times cannot be considered as premium services. These should be provided to all clients if there is need. This may raise serious ethical concerns. Please revise the sentence and clearly define what you meant by premium service.

Line 162: the range 18,500NGN – 100,000NGN is very wide and may mask some details. Would you to base this categorization on minimum wage, average wage and above average based on Nigeria context instead of just for convenience?

Lines 196 -7: Figures should be represented at the end while tables should come after the 1st paragraph where they are mentioned.

Lines 203 – 10: You presented WTP for discrete services only. Will be good to know how many clients will be willing to pay for all the services. Additionally, information about clients willing to pay for consultation, ARV and VL or VL and ARV only e.t.c. may be helpful in prioritizing which service combination will be most affordable. Additionally, the authors didn’t establish the standard cost of the services, or are the initiated prices in Fig 1 the standard costs of these services? For example, what will be the market value of a month-dose of ARVs in the study states? Even though in line 280 4.85USD has been quoted as monthly cost of TLD, is this market price in the study states or subsidized price being supplied to the SFI sites?

Line 295: It will be good to know how many total clients are receiving care in the 92 selected sites and what proportion is the 1781 clients. How the sample size of 1781 clients were determined. This is important for the inference made in line 295 -6. Perhaps you can elaborate on the appropriate section in your methods.

Line 324: better to state that the contents are the responsibility of the authors and not FHI360.

6. PLOS authors have the option to publish the peer review history of their article (what does this mean?). If published, this will include your full peer review and any attached files.

Reviewer #1: No

Reviewer #2: No

---

## [Author Response · Author response to Decision Letter 0]

12 Jul 2023

July 10, 2023

Dr. Ibrahim Jahun,

Academic Editor,

PLOS ONE

Re: Manuscript number - PONE-D-23-14068

Thank you for considering and reviewing our manuscript entitled “Willingness to pay for antiretroviral therapy, viral load and premium services; A contingent valuation survey of people living with HIV in southern Nigeria”. We appreciate the opportunity to revise the manuscript and resubmit to your journal.

We have revised the manuscript as requested by the reviewers and prepared responses to comments as described in detail below.

Reviewer 1 comments

1. Comment: 

Please have a copy editor review the paper. There are incomplete sentences and misspelled words throughout the paper and figures. I suggest adding place and date to figure and table titles.

Response:

Thank you very much for the comments. We have taken the time to review the manuscript and corrected all identified issues. We also included the “year” and place in the titles for the tables and figures. For example, Table 1 now reads: “Table 1: Study sample characteristics and willingness to pay for consultation, ARVs, HIV viral load testing and premium ART services: Willingness to Pay for HIV services survey 2019 – Nigeria”

2. Comment:

Readers not familiar with Naira could benefit from providing the exchange rate relative to more commonly traded currencies, i.e., dollar, euro, or pound. Authors should be consistent with currency used. In one section both Naira and dollars were used.

Response:

We provided the exchange rate in lines ‘171 – 2’ which reads “The study assessed the US dollar value of the maximum amount PLHIV were willing to pay for ART services by converting at a rate of 1$ to NGN359.98 which was prevalent at the time of the study.” We also included USD equivalents for all results related to our research questions to aid readability.

Reviewer 2 comments

1. Comment: 

The manuscript is technically sound requiring minor edits as outlined below. Additionally, the authors should adhere to PLOS ONE guidelines on tables and figures.

Response:

Thank you for your review. We have edited the tables and figures to be more consistent with the PLOS ONE guidelines as indicated. 

2. Comment: 

Lines 80 – 2: The goal is not to only attain UNAIDS targets but to attain HIV epidemic control. Consider adding this caveat.

Response:

We included this caveat. Lines 81 - 2 now read – “However, this success, amidst reduced from the donors, requires the country to develop a plan to maintain the momentum towards reaching and maintaining the UNAIDS targets for HIV epidemic control.” 

3. Comment: 

Lines 146 – 7: pharmacy, lab services and longer consultation times cannot be considered as premium services. These should be provided to all clients if there is need. This may raise serious ethical concerns. Please revise the sentence and clearly define what you meant by premium service.

Response:

This is an important point. We clarified the intention behind the statement. Our intention is not to suggested that pharmacy and laboratory services are exclusively premium. Lines 148 - 9 now read – “To assess the amount participants were willing to pay for clinical consultation, ARVs, Viral Load (VL) tests and premium services which could include fast-track clinic, concierge pharmacy and laboratory services (where services are delivered to clients in the fast-track clinic), flexible ART scheduling and longer clinic consultation time.” 

4. Comment: 

Line 162: the range 18,500NGN – 100,000NGN is very wide and may mask some details. Would you to base this categorization on minimum wage, average wage and above average based on Nigeria context instead of just for convenience? 

Response:

Thanks for the observation. Unfortunately, this data was collected as categorical data which we are unable to recategorize post-hoc. We created 3 categories during the survey as NGN18,500 or less which represented minimum wages while we considered NGN100,00 or more as at least upper middle class or better (at the time). The wide range between NGN18,500-100,000 is acknowledged as a potential limitation. 

5. Comment: 

Lines 196 -7: Figures should be represented at the end while tables should come after the 1st paragraph where they are mentioned. 

Response:

We have moved the tables and figures to be appropriately situated based on the PLOS ONE guidelines. 

6. Comment: 

Lines 203 – 10: You presented WTP for discrete services only. Will be good to know how many clients will be willing to pay for all the services. Additionally, information about clients willing to pay for consultation, ARV and VL or VL and ARV only etc. may be helpful in prioritizing which service combination will be most affordable. 

Response:

Thanks for the suggestion. We agree that presenting willingness to pay for all ART services would be useful information, but we are reluctant to extrapolate from the data given that we did not directly ask this question. For example, given the cost implications, it would be wrong for us to suggest that if a respondent said yes to all services in isolation, it means they would say yes to all services together when this might imply more costs than when considered in isolation. We have added phrasing in the discussion of limitations and highlight other studies that have answered that specific question. Lines 324 - 6 now read “As this study was expected to identify possible areas whose costs may be transferred from the donor to the clients, questions were only asked about willingness to pay for the different Antiretroviral therapy (ART) service delivery areas. Additional studies may go further to explore the willingness to pay for the comprehensive ART services.” Regarding the standard cost of services, attempt was made to establish these costs informally as detailed below:

ARV Cost – This was arrived @ based on the 30-day supply of GF TLD PPM cost and the anticipated distribution cost. These drugs are distributed free with donations from international donors (USG/GF) and not currently being sold in supported facilities, so we are unable to determine the actual cost per state.

Consultation cost – This was based on the average charge for non –HIV related outpatient consultation in our supported facilities.

Viral Load Cost- Although this was provided for free across our supported facilities using public labs, this was the minimum possible charge for provision of this test in our supported private facilities.

Premium Services – This was arrived at based on the average cost of specialist consultation in our supported private facilities. 

7. Comment: 

Line 295: It will be good to know how many total clients are receiving care in the 92 selected sites and what proportion is the 1781 clients. How the sample size of 1781 clients were determined. This is important for the inference made in line 295 -6. Perhaps you can elaborate on the appropriate section in your methods. 

Response:

How the sample size was calculated has been added to the method section (Lines 129-133). Please note that 1,740 was calculated as minimum sample size but 1,781 survey responses were collected and analyzed. The sampling frame shows that 68,570 clients were receiving care from the selected facilities. 

8. Comment: 

Line 324: better to state that the contents are the responsibility of the authors and not FHI360. 

Response:

This correct. We have adapted the acknowledgments section. Lines 340 - 2 now read “This study was made possible by the generous support of the American people through the U.S. Agency for International Development (USAID). The contents are the responsibility of the authors and do not necessarily reflect the views of Family Health International (FHI 360) and USAID.” 

Again, we appreciate the detailed review and look forward to hearing from you.

Sincerely,

Olusola Joseph Sanwo

MBBS (Ibadan), MPH (Harvard)

---

## [Editor Report · Decision Letter 1]

20 Jul 2023

Willingness to pay for antiretroviral therapy, viral load and premium services; A contingent valuation survey of people living with HIV in southern Nigeria

PONE-D-23-14068R1

Dear Dr. Sanwo,

We’re pleased to inform you that your manuscript has been judged scientifically suitable for publication and will be formally accepted for publication once it meets all outstanding technical requirements.

Kind regards,

Ibrahim Jahun, MD, MSC, PhD

Academic Editor

PLOS ONE

---

## [Editor Report · Acceptance letter]

1 Aug 2023

PONE-D-23-14068R1 

Willingness to pay for antiretroviral therapy, viral load, and premium services; A contingent valuation survey of people living with HIV in southern Nigeria 

Dear Dr. Sanwo:

I'm pleased to inform you that your manuscript has been deemed suitable for publication in PLOS ONE. Congratulations! Your manuscript is now with our production department. 

Kind regards, 

on behalf of

Dr. Ibrahim Jahun 

Academic Editor

PLOS ONE